# Assessment of Indoor Air Quality in Residential Buildings of New England through Actual Data

**Fernando del Ama Gonzalo** [1,*] **, Matthew Griffin** [1]**, Jacob Laskosky** [1]**, Peter Yost** [2]
**and Roberto Alonso González-Lezcano** [3]

1    Department of Sustainable Product Design and Architecture, Keene State College, University System of New Hampshire, Keene, NH 03435, USA; Matthew.Griffin@keene.edu (M.G.); Jacob.Laskosky@keene.edu (J.L.)
2    Building-Wright, Brattleboro, VT 05301, USA; peter@building-wright.com
3    Escuela Politécnica Superior, Montepríncipe Campus, Universidad San Pablo-CEU, CEU Universities, 28668 Madrid, Spain; rgonzalezcano@ceu.es
*    Correspondence: fernando.delama@keene.edu

**Abstract:** Several studies on indoor air quality (IAQ) and sick building syndromes have been completed over the last decade, especially in cold countries. Efforts to make homes airtight to improve energy efficiency have created buildings with low ventilation rates, resulting in the build-up of indoor pollutants to harmful levels that would be otherwise unacceptable outdoors. This paper analyzed the infiltration rates, indoor temperatures, and variations in $CO_2$, 2.5 μm particulate matter ($PM_{2.5}$), and total volatile organic compound (TVOC) concentrations over the fall of 2021 in several homes in New England, USA. A relationship between outdoor and indoor conditions and ventilation strategies has been set using the results from blower door tests and actual indoor air quality data. Although all case studies lacked mechanical ventilation devices, such as those required by ASHRAE Standard 62.2, natural ventilation and air leakage have been enough to keep VOCs and $PM_{2.5}$ concentration levels at acceptable values most of the studied time. However, results revealed that 25% of a specific timeframe, the occupants have been exposed to concentration levels of $CO_2$ above 1000 parts per million (ppm), which are considered potentially hazardous conditions.

**Keywords:** indoor air quality; air tightness; residential buildings; environmental quality conditions

## 1. Introduction

Indoor air quality (IAQ) is determined by the presence of pollutants and thermal-humidity conditions in the indoor environment that may negatively affect building occupants' health, comfort, and performance [1]. Indoor air quality (IAQ) is one of the essential criteria for evaluating the quality of a building, according to the U.S. Environmental Protection Agency [2]. At increased concentrations and exposure times, these pollutants harm users' health. Several studies performed on indoor air quality (IAQ) have shown that the number of people present in a room, physical activities, and combustion of solid fuels raise carbon dioxide levels [3]. After implementing energy retrofits, which are mainly focused on improving the thermal parameters of the building envelope, the concentration of $CO_2$ in the indoor environment increases [4]. Therefore, improving the airtightness makes the problem of $CO_2$ concentration more relevant [5,6]. Besides, low outdoor ventilation rates have led to the accumulation of indoor pollutants to dangerous levels that would be otherwise unacceptable outdoors, especially in cold countries [7–9]. According to some studies, indoor air can be up to five times more polluted than outdoor air [10]. Homes are not designed for intense use and, sometimes, lack appropriate ventilation strategies and indoor air renewal [11]. Since household users are exposed to several contaminants, such as chemicals for cleaning, animal fur, and tobacco smoke [12,13], natural ventilation is an effective means to reduce those contaminant agents. However, pollution from outside, such as pollen, dust, and industrial and traffic emissions, must also be considered. Research on

the indoor air quality in homes during the COVID-19 lockdown is limited, although some studies highlight that the air quality has decreased due to cleaning products and a lack of ventilation [14].

When the homes selected for this research were originally constructed, few demands in terms of airtightness were placed on the envelope, windows, and doors. Therefore, the exterior air enters the interior through window joints and leaks around openings. In this way, natural ventilation and a sufficient supply of fresh air to the interior are not guaranteed without a certain degree of self-discipline from building occupants to ensure ideal indoor air quality conditions. For example, if a building is not adequately ventilated, moisture in the dwelling can increase, allowing mold and other fungi to grow [15]. Other components that make up a comprehensive strategy for indoor air quality include limiting materials and activities that provide the source of pollutants and employing local exhaust in kitchens and bathrooms where high concentrations of contaminants are likely to occur [16–18]. The relationship between airtightness and air change rates should be analyzed in cold countries, as indoor air quality, the comfort of residents, and energy efficiency are affected by the examined variables.

In New England, ventilation and infiltration heating loads contribute between 20% and 40% to the total losses accounted for [19,20]. The air moves into and out of a building through pressure differentials in the building envelope. Unintentional leakages are referred to as infiltration, depending on the size and distribution of air leakage pathways and the pressure difference between indoors and outdoors [21]. Ventilation strategies have different connections with air leakage. Since air infiltration is unintentional, quantifying its contribution is essential in the design phase of a ventilation system [22]. Over the last few years, the main effort has been put into energy savings. Therefore, in cold climates, the lack of suitable ventilation has led to higher indoor pollutant levels. As a result, the average daily concentration of particulate matter, especially those with a diameter of 2.5 μm or smaller, $PM_{2.5}$, increased 12% and the mean total volatile organic compound TVOC concentration increased from 37% to 56% [23,24].

Occupant activities or materials and furnishings in the building can produce indoor contaminants. The indoor air pollutants most commonly associated with building materials and building-related activities are formaldehyde and volatile organic compounds (VOCs). In contrast, indoor combustion presents a significant source of VOCs and particles [25]. Mechanical ventilation systems or infiltration can introduce pollutants in indoor air, such as $NO_x$, $SO_x$, and ozone, dust, allergens, molds, and toxins [26]. Besides those pollutants, radon is also an indoor air contaminant from outdoors [27].

This paper describes a comparative survey of air quality in residential buildings in New England, more specifically, southwestern New Hampshire and central Connecticut. During this study, the authors measured several parameters to determine the indoor air quality of homes; these variables include indoor temperature, relative humidity, variations in $CO_2$, 2.5 μm particulate matter ($PM_{2.5}$), and total volatile organic compound (TVOC) concentrations. Firstly, blower door tests were completed in all case studies to assess the airtightness. Secondly, IAQ parameters monitored over fall 2021 were analyzed and compared with the occupants' habits and schedules. Finally, correlation between indoor and outdoor conditions, airtightness, and indoor pollutant levels were analyzed. This paper aimed to analyze indoor air quality parameters, such as $PM_{2.5}$, $CO_2$, and TVOC, in four representative case studies in New England to correlate different parameters to occupant behavior. The concentrations of $CO_2$, $PM_{2.5}$, and TVOCs were measured over two months concerning the thermal-physical parameters of indoor air quality to identify health risks of the occupants due to this constant exposure and the regular cycles in the dwellings.

## 2. Materials and Methods

Four single-family houses were selected for the cross-state observation: New Hampshire and Connecticut. The four case studies are representative of different New England

typologies, mainly lightweight construction without heavy brick or concrete bearing walls. The case studies are detached houses with four facades exposed to the wind.

### 2.1. Analysis of the Case Studies

New Hampshire has a humid continental climate. In this region, the winters are long and cold, and heavy snow is usual. The summer months are moderately warm, with a rainfall peak in July. Northern Connecticut presents a hot-summer version of the humid continental climate. Winters are shorter, with less snowfall, and summers can occasionally be hot and humid, with high temperatures (up to 30 °C). Convective thunderstorms are common in these months, some of which can become severe, although annual rainfall is spread evenly throughout the year.

When it comes to the residential typologies analyzed in this paper, the household stock shared many building and behavioral characteristics with other northeastern US suburban areas. The most widespread residential buildings are single-family dwellings [28,29]. Therefore, four case studies representative of the typical housing were selected for monitoring their environmental data under actual operational conditions over fall 2021. Figure 1 illustrates the selected case studies. CS1 and CS2 are located in Keene, New Hampshire, USA. The first is a prefabricated mobile home; the second is a traditional New England typology built in the late XIX century that has been turned into a facility available for rent for students' off-campus needs. CS3 and CS4 are located in Plantsville and New Britain, respectively. Both are representative typologies of the residential construction in Connecticut.

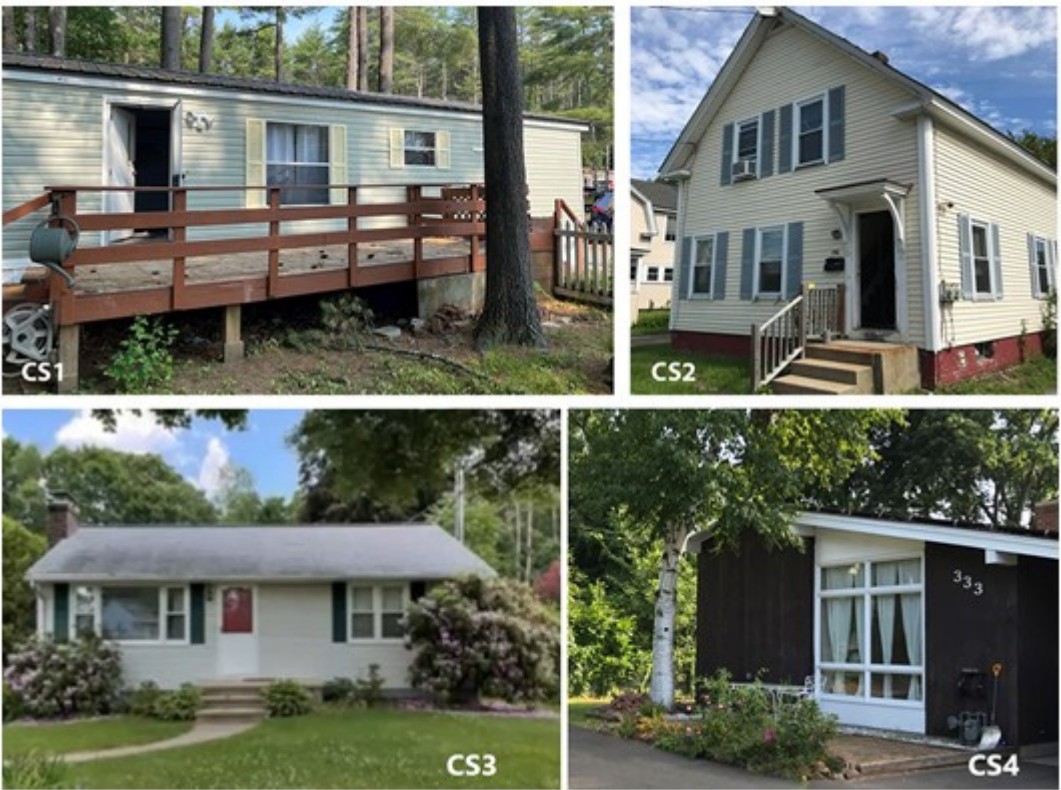

**Figure 1.** Case studies. **CS1** and **CS2** are located in New Hampshire. **CS3** and **CS4** are located in Connecticut.

The case studies were selected due to the features of the house and the family composition to present a sample of the most common types of families: from a single adult living alone (CS1), which represents 25.3% of the population of New England, to off-campus housing for students (CS2), and families with young adult children (CS3, CS4). Table 1 shows the features of the case studies, along with information about heating, ventilation,

and air conditioning (HVAC) systems and occupants' habits and schedules. CS1 is a mobile home, meaning that it is a prefabricated structure built in a workshop on a permanently attached chassis. CS1 was built according to the Federal National Manufactured Housing Construction and Safety Standards Act of 1974. The exterior cladding of the home consists of vinyl siding. The glazing used in the home consists of single glass panes with aluminum frames. CS2 is a traditional New England house built at the end of the 19th century with a balloon frame structure and wood siding. The windows in this home are made of single-pane glazing and wood frames. CS3 is a typical ranch house built during the 1950s. It is built using platform wood framing with roof trusses and is clad in vinyl siding. The windows in this home utilize double-pane glazing with PVC frames. Finally, CS4 was built in the 1970s with a unique architectural design, including a centralized double volume space connecting the living room, dining room, and kitchen. In recent years, this house underwent an energy retrofit, including replacing single glazing and wooden frames with triple glazing with airtight PVC frames.

**Table 1.** Case study features.

| Case Study | CS1 | CS2 | CS3 | CS4 |
|---|---|---|---|---|
| Construction year | 1995 | 1890 | 1955 | 1979 |
| Number of occupants | 1 | 4 | 3 | 2 |
| Smokers | No | Yes | Yes | No |
| Pets | No | No | Yes | No |
| Floor area ($m^2$) | 79 | 210 | 120 | 164 |
| Indoor volume ($m^3$) | 281 | 656 | 381 | 418 |
| Height (m) | 2.62 | 5.6 | 3.8 | 4.5 |
| Window area ($m^2$) | | | | |
| DHW system | Electric | Gas | Gas | Gas |
| Heating system | Gas furnace and air ducts | Gas furnace and air ducts | Gas furnace and air ducts | Gas boiler and radiators |
| Cooling system | No | No | Heat pump and air ducts | No |
| Ventilation system | Natural ventilation operable windows | Natural ventilation operable windows | Natural ventilation operable windows | Natural ventilation operable windows |
| Stove or fireplace | No | No | Yes | Yes |
| Carpets | No | No | Yes | No |
| House Typology | Single family | Single family | Single family | Single family |
| Floors | 1 | 2 + basement | 2 | 2 + basement |
| Facades exposed to the wind | 4 | 4 | 4 | 4 |

## 2.2. Monitoring

Thermal ambient monitoring was carried out following ISO 23210 specifications, both instrumentation and methods [30]. The monitoring devices were in constant operation over the tested period. Infrared thermal cameras were used to record periodic contrast measurements in situ to ensure the consistency and accuracy of the measurements of the monitoring systems [31–33]. Regarding indoor air quality, the most accepted standard for housing in the United States is ASHRAE Standard 62.2, "Ventilation and Indoor Air Quality in Low-Rise Buildings" [34]. It mainly relies on mechanical ventilation to introduce outside air and decrease indoor pollutants. In addition, there are other requirements, such as particulate filtration and exhaust-based ventilation in kitchens and bathrooms [35]. Ventilation has progressed from the primary concerns about condensation and mold prevention to addressing the issue of harmful indoor air pollution. Most recently, ventilation has been considered a factor in reducing virus transmission. Indoor air pollution sources are widespread and vary dramatically from house to house. Sources of pollutants include cooking, cleaning, fires, candles, and even building and decorating materials. Experts agree that both source removal and ventilation are essential for reducing indoor air pollutants. If the property is

located near a busy road, ventilation has to be combined with filtration. ASHRAE standard describes the functions and minimum specifications for mechanical and natural ventilation systems and infiltration rates to produce acceptable IAQ in houses. Addressing airtightness is a critical problem both for existing and new buildings. In situ blower door tests are the most common way of assessing building airtightness [36,37]. Firstly, temperature, humidity, carbon dioxide ($CO_2$), $PM_{2.5}$, and VOCs levels were analyzed in 4 housing units. Secondly, relationships between airtightness and $CO_2$ concentration were examined in this research. The AWAIR home sensor is a small device that has a collection of sensors that allows users to measure different IAQ variables [38,39]. These devices were deployed in each dwelling unit for several weeks and placed in key locations to ensure that the validity of the data would not be compromised.

### 2.3. Exterior Ambient Quality

While conducting this research it was important to understand how the exterior atmospheric air behaved during the testing period. While a detailed analysis is preferred, it is out of the scope of this article and, therefore, a general analysis and understanding of the exterior conditions will suffice. Primary sources of pollutants included traffic and heating systems of nearby buildings. The Air Quality Index (AQI), defined by the Environmental Protection Agency (EPA) [2], is a standardized system that is used to report pollution levels in the atmosphere. This AQI allows citizens to understand the potential health issues regarding high levels of pollutants. The AQI is converted to a value between the range of 0 and 500 for ease of understanding, with 0 being the cleanest air, while 500 is the zenith of air pollution levels. There are five major air pollutants considered by the EPS to set the AQI index: ground level ozone, particle matter pollution, CO, $SO_2$, and $NO_2$ [40]. Figure 2a shows outdoor levels of $PM_{2.5}$, $PM_{10}$, $SO_2$, and $NO_2$ over 2021 in Keene, New Hampshire. Average particle matter levels remained below 10 $\mu g/m^3$, with only 5% of days reaching values above 20 $\mu g/m^3$, so the AQI index can be considered as very good. When it comes to $NO_2$, there were excellent values between 10 and 20 parts per billion (ppb) 72% of days over the year. EPS has set an acceptable 1 h $NO_2$ standard at the level of 100 ppb and an annual average $NO_2$ standard of 53 ppb. Sulfur dioxide ($SO_2$) levels also remained far below the values established by the EPA regulatory standard of 75 parts per billion (ppb). Figure 2b shows outdoor levels of $PM_{2.5}$, $PM_{10}$, $SO_2$, and $NO_2$ over 2021 in Hartford, Connecticut. This area is far more populated than the studied area in Keene, New Hampshire, with higher levels of road traffic and industry. $NO_2$ values ranged between 30 and 40 parts per billion (ppb) 60% of days over the year, with peaks of 50 ppb. Nevertheless, the levels were below the recommended annual average $NO_2$ standard of 53 ppb.

### 2.4. Blower Door Test

The air permeability of the envelope can lead to different ventilation strategies [41]. A blower door is a powerful fan that mounts into the frame of an exterior door. The fan pulls air out of the house, forcing air through cracks and small penetrations in the structure's thermal envelope. The building envelope and its materials and components are designed to have a long service life. However, it is unknown if PE foil and tapes used in residential construction have a sufficient service life, as many tested taped assemblies only lasted about 15–20 years. Therefore, four blower door tests were performed in all the tested houses to assess the amount of exterior air infiltration and the influence of infiltration on the indoor air renovation. A reference air pressure difference of 50 Pa was set to assess the airtightness of the case studies. Building volumes were used as size scaling to provide air changes per hour (ACH) at 50 Pa, which is a suitable magnitude for detached single-family houses. A range of $ACH_{50}$ values between 4 $h^{-1}$ and 8 $h^{-1}$ is considered acceptable for natural ventilation strategies, whereas homes with $ACH_{50}$ values below 1 $h^{-1}$ require mechanical ventilation with heat recovery devices [42]. According to ASHRAE 90.2 [43], low-rise residential buildings' maximum envelope air leakage should be no greater than 5 $ACH_{50}$ in climate zones 0, 1, and 2, and no greater than 3 $ACH_{50}$ in climate zones 3 through

8. For example, the Hartford area in Connecticut belongs to climate zone 5–A (cool humid), and Keene, New Hampshire, belongs to climate zone 5–C (cold humid). Therefore, the value of $ACH_{50}$ in all the cases should not be greater than 3.

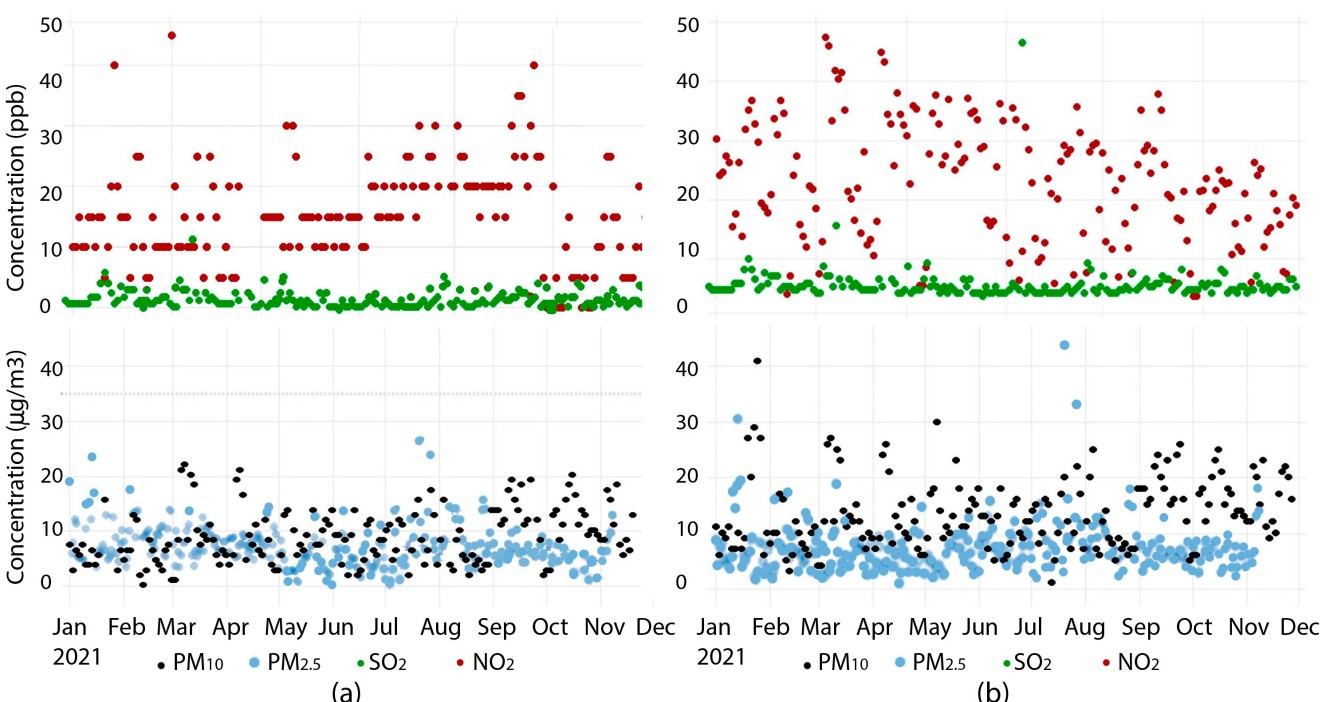

**Figure 2.** Outdoor levels of $PM_{2.5}$, $PM_{10}$, $SO_2$, and $NO_2$ over 2021 in (**a**) New Hampshire and (**b**) Connecticut. Data from [40].

## 3. Results

Before analyzing the parameters that define indoor air quality, blower door tests were completed in each house to assess their airtightness. Since none of the case studies were equipped with mechanical ventilation, air infiltration might be the only source of fresh air, especially when the weather gets cold at the end of the fall.

### 3.1. Blower Door Test Results and Ventilation Requirements

Blower door tests have been carried out in all case studies. ASHRAE standards have been applied to determine the required amount of natural ventilation [32]. Equation (1) shows the total ventilation rate required by ASHRAE in international units.

$$Q_{tot} = 0.15 A_{floor} + 3.5(N_{br} + 1), \tag{1}$$

where $Q_{tot}$ is the required ventilation rate in L/s, $A_{floor}$ is the floor area in m$^2$, and $N_{br}$ is the number of bedrooms in the dwelling unit. If a blower door test has been performed, then the expression that shows the required mechanical ventilation rate $Q_{fan}$, in liter per second, is shown in Equation (2):

$$Q_{fan} = Q_{tot} - \Phi \left( Q_{inf} \ x \ A_{ext} \right) \tag{2}$$

being:

$$\Phi = \frac{Q_{inf}}{Q_{tot}}, \tag{3}$$

where $A_{ext}$ is 1 for detached dwelling units. $Q_{inf}$ is the infiltration rate in L/s, and its value is shown in Equation (4).

$$Q_{inf} = 0.052 \ x \ Q_{50} \ x \ wsf \ x \ (H/H_r)^z \tag{4}$$

where $Q_{50}$ is the leakage rate at 50 Pa, *wsf* is the weather and shielding factor, $H$ is the vertical distance between the highest and lowest above-grade points, $H_r$ is the reference height, 2.5 m, and z is 0.4. The values for wsf were 0.51 in Keene, NH, and 0.5 in Hartford, CT, according to ASHRAE Standard 62.2 Appendix A [32].

Table 2 shows the results, as well as a chart outlining $ACH_{50}$ values and their corresponding attributes. CS1 had an $ACH_{50}$ value of 5.21, with CS2 having a value of 8.04, CS3 having a value of 5.54, and, finally, CS4 having a value of 1.71. In this case, the best-performing home was CS4 with a value of 1.71 $ACH_{50}$, as was expected after the energy retrofit, while the worst performing home was CS4 with an 8.04 $ACH_{50}$. CS4 turned out to be a very leaky building, given that the structure was built during the 19th century. During this time, airtightness was likely one of the last things that home builders were concerned with.

**Table 2.** Required ventilation rate ($Q_{fan}$) in the houses.

| Case Study | Volume (m$^3$) | $N_{br}$ | $ACH_{50}$ | $Q_{50}$ (L/s) | *wsf* | $H/H_r$ | $Q_{tot}$ (L/s) | $Q_{inf}$ (L/s) | $Q_{fan}$ (L/s) |
|---|---|---|---|---|---|---|---|---|---|
| CS1 | 281 | 2 | 5.21 | 408 | 0.51 | 1.04 | 22 | 11 | 17 |
| CS2 | 656 | 4 | 8.04 | 1465 | 0.51 | 1.22 | 49 | 42 | 13 |
| CS3 | 381 | 3 | 5.54 | 587 | 0.50 | 2.23 | 29 | 21 | 13 |
| CS4 | 418 | 2 | 1.71 | 199 | 0.50 | 1.60 | 35 | 6 | 34 |

According to ASHRAE 90.2 [43], only CS4 met the requirements for climate zone 5, with lower than 3. Because of the high airtightness, this house should have a mechanical ventilation system with heat recovery. CS1 and CS3 showed average values for residential construction in the area and are suitable for natural ventilation strategies.

*3.2. Temperature and CO$_2$ Concentration Analysis*

A range from 250 to 400 ppm in $CO_2$ concentration in outdoor ambient air is considered acceptable, whereas concentrations above 1000 ppm are regarded as poor indoor air quality and can cause drowsiness. Figure 3 illustrates $CO_2$ concentrations in New Hampshire and Connecticut case studies. In both cases, $CO_2$ concentrations have shown wide fluctuations over time. CS1 concentrations range from 400 ppm (barely equivalent to outdoor concentration) to 900 ppm. CS2 was the case with the lowest volume per occupant ratio, which led to high concentrations of $CO_2$, with peaks above 1500 ppm. Nevertheless, the average $CO_2$ level was within the acceptable threshold of WHO standards. CS1 temperature was often below comfort due to the single person's schedule, who was only at home in the morning and the evening. The temperature in CS2 was often above comfort levels (25 °C). CS2 had the heating included in the rent, so the occupants did not care about overheating.

Concentration level in CS4 hardly ever surpassed 1000 ppm. In contrast, CS3 showed high levels in $CO_2$ concentrations, with consistent values above 1400 ppm and peaks of 2000 ppm. Concentrations above 2000 ppm can lead to loss of attention, nausea, and increased heart rate [43]. Over two months, the average $CO_2$ concentration was 647 ppm in CS3, and 1000 ppm was the 87th percentile.

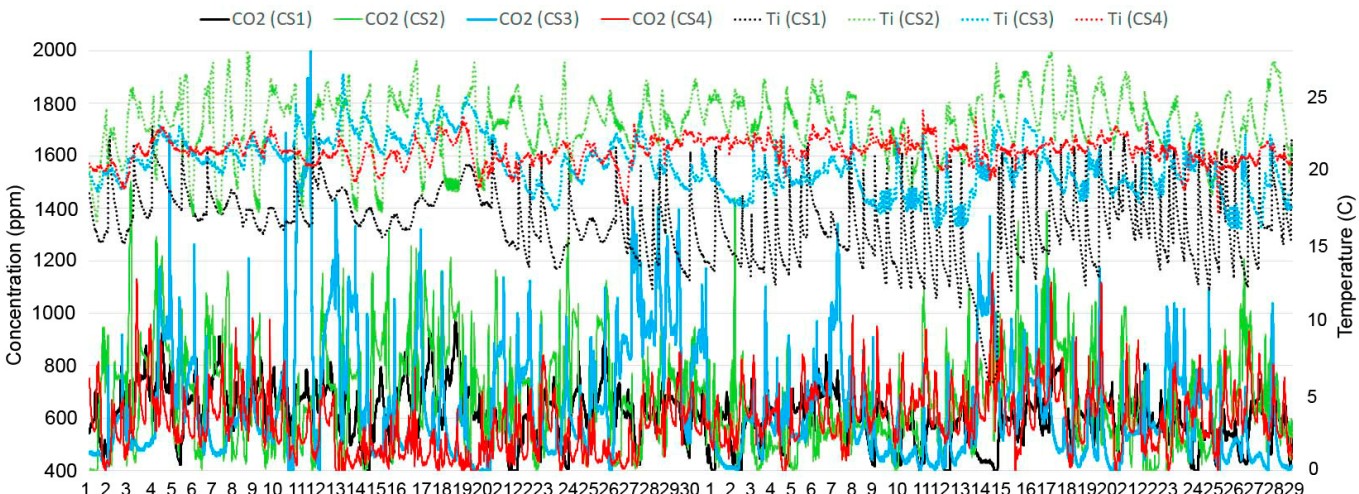

**Figure 3.** Indoor temperature and $CO_2$ concentration in New Hampshire (CS1, CS2) and Connecticut (CS3, CS4) case studies from 1 October 2021 to 29 November 2021.

### 3.3. PM$_{2.5}$ and TVOC Concentration Analysis

The outdoor concentration of PM$_{2.5}$ in both locations did not present significant variations throughout the analyzed period, except for specific peaks, with a daily average of around 7 µg/m$^3$. An acceptable limit of PM$_{2.5}$ concentration is 10 µg/m$^3$, and a dangerous threshold is set at 25 µg/m$^3$ by the World Health Organization (WHO) [44,45]. Figure 4 shows outdoor concentrations of PM$_{2.5}$ and ozone in both locations over October and November 2021. The tropospheric urban ozone levels were low to very low over the analyzed period. Average ozone level values of 25.31 ppb and 22 ppb were measured in Hartford and Keene areas, respectively. Peaks of 50 ppb were far from the recommended 80ppb, eight-hour average EPA's National Ambient Air Quality Standard for ozone. Average particle matter levels remained between 3 µg/m$^3$ and 10 µg/m$^3$, with only 3 days reaching values above 18 µg/m$^3$.

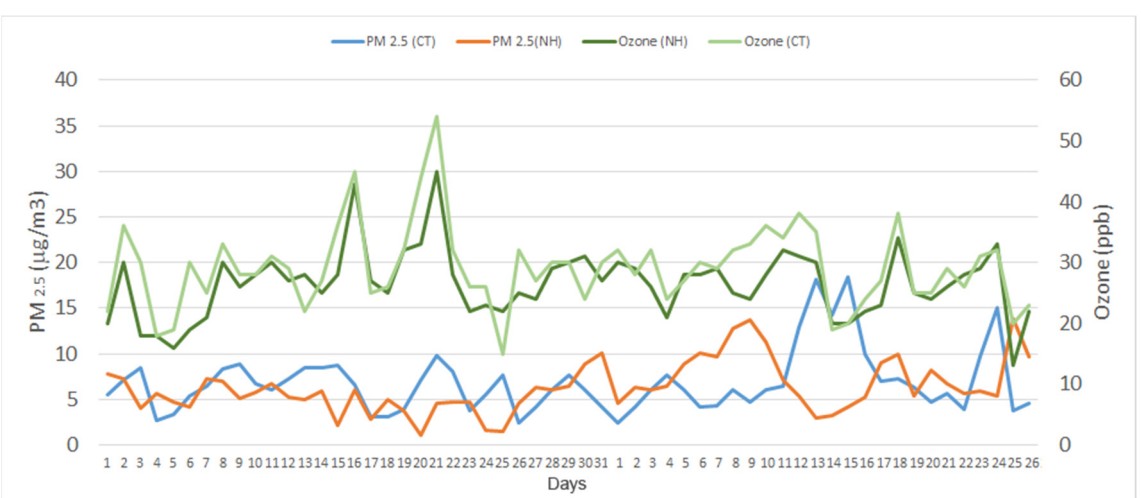

**Figure 4.** Outdoor levels of PM$_{2.5}$ and ozone in both locations, Connecticut (CT) and New Hampshire (NH), from 1 October 2021 to 29 November 2021. Data from [40].

Several internal sources can be regarded as particle emitters, which can assume a relevant role in high-intensity indoor spaces, such as student dorms. For example, cooking without a proper extraction system and cleaning products can cause the emission of vapor and aerosols. Figure 5 illustrates that indoor PM$_{2.5}$ and VOC averages of both locations

were higher than the city reference outdoor values. PM$_{2.5}$ indoor concentrations did not show a clear pattern. Average values through the monitoring period were 3.01 µg/m$^3$ (CS1), 16.9 µg/m$^3$ (CS2), 3.37 µg/m$^3$ (CS3), and 9.59 µg/m$^3$ (CS4). CS2 surpassed the recommended annual threshold for PM$_{2.5}$ concentration of 10 µg/m$^3$, with peaks of 880 µg/m$^3$. The threshold of 10 µg/m$^3$ corresponded to a 77th percentile.

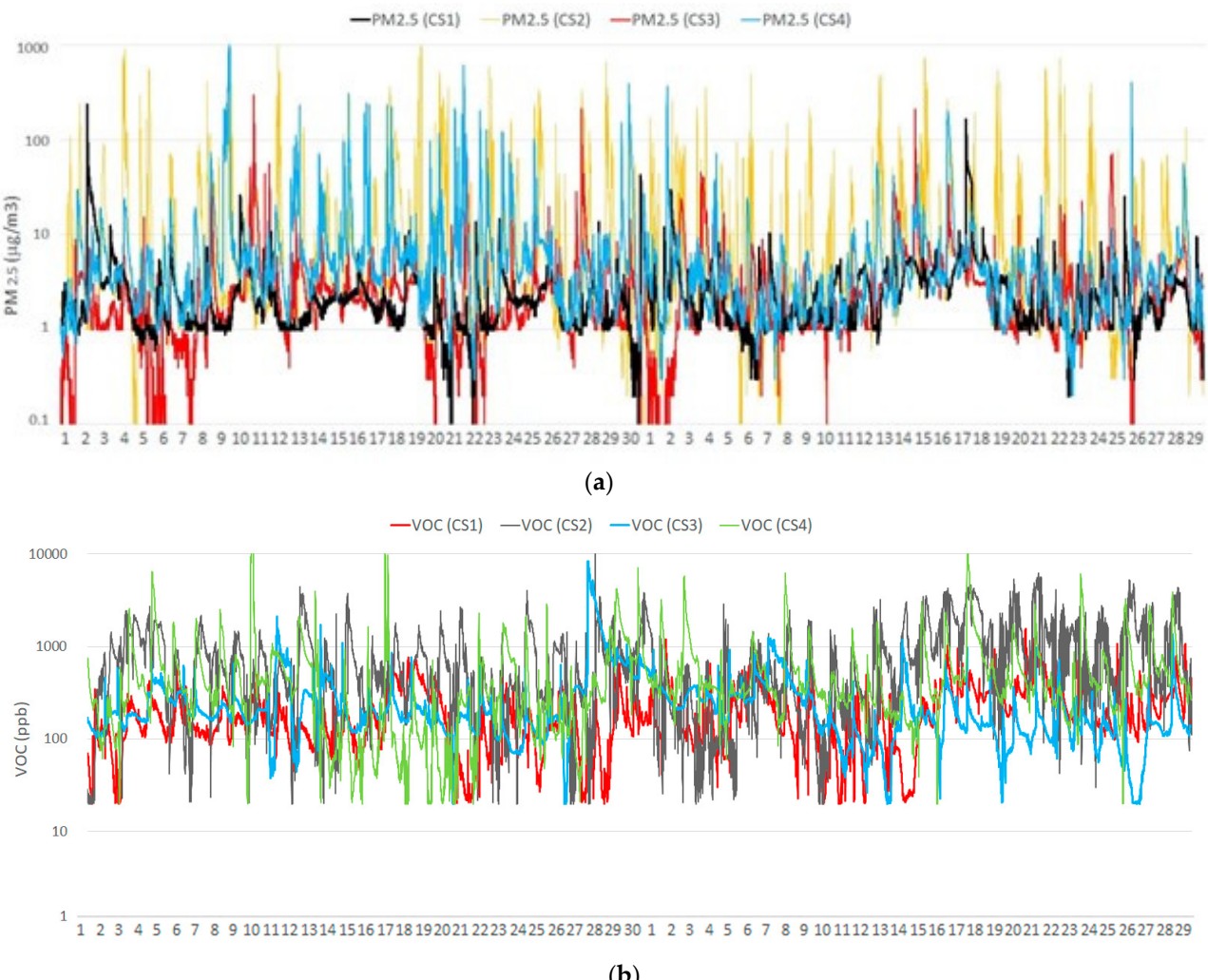

**Figure 5.** Description of PM$_{2.5}$ concentration levels (**a**) and VOC concentration levels (**b**) in New Hampshire (CS1, CS2) and Connecticut (CS3, CS4) case studies from 1 October 2021 to 29 November 2021. Logarithmic scale.

CS4 average concentration was close to 10 µg/m$^3$, with this threshold as the 86th percentile. The lowest peak value was in CS1 with 199 µg/m$^3$, followed by CS3 with 235 µg/m$^3$.

When it comes to TVOC concentrations, there was a considerable impact from indoor materials, generating an almost constant base level of 50 ppb. Occupants' movement and activity can increase the effect of the blend of indoor contaminants, with a significant impact on the recorded values. There was also a relevant contribution of the use of daily housework products and personal care. The reduction in TVOC concentration depended on the ventilation habits, for an external contribution of VOCs is not usual. VOC concentration values below 120 ppb were considered appropriate for a healthy environment, whereas values above 1200 ppb could provoke health risks. In all cases, indoor average TVOC concentration was below 1200 ppb. The average values were 208 ppb for CS1, 858 ppb for CS2, 286 ppb for CS3, and 606 ppb for CS4, without discernible patterns, except for CS4,

where the cooking habits of the occupants show peaks of 9000 ppb around 7:00 p.m. on a daily basis.

## 4. Discussion

This section discussed the results shown in the previous section focusing on the correlation between indoor and outdoor PM values, and between all indoor air quality indicators.

### 4.1. Correlation Matrix

As the weather got cold, natural ventilation was restricted to short periods, resulting in two different indoor/outdoor environments where air interchange mainly depended on airtightness. On the other hand, when the weather was mild, building enclosure permeability increased due to the opening of the windows. The first goal of this section was to analyze the correlation between indoor and outdoor $PM_{2.5}$ levels in all case studies ($PM_{2.5}$ (IN)/$PM_{2.5}$ (OUT)). Table 3 shows that, despite the expected correlation between indoor and outdoor $PM_{2.5}$ levels, the only case study showing a significant relationship (0.47) was CS2 due to the leaky airtightness of the house ($ACH_{50} = 8.04$). In the rest of the cases, indoor sources of particle emissions are far more critical than outdoor ones. CS4 showed the lowest correlation between indoor and outdoor $PM_{2.5}$ concentrations due to the highest airtightness level. When analyzing the correlation between IAQ parameters, there was a high correlation value between $CO_2$ levels and VOCs (0.542) in CS2, related to indoor home activities and a lack of natural ventilation. Likewise, in CS3, there was a significant correlation between indoor temperature, $CO_2$ levels, and VOCs (0.456). The internal production of VOCs can be explained by human activities in the kitchen, where the fireplace can cause high levels of $CO_2$ and VOCs. The reduction in natural ventilation rate revealed a strong inverse correlation to outdoor temperature and indoor air quality indicators.

**Table 3.** Correlation matrix between IAQ parameters in the case studies.

|  | (CS1) | (CS2) | (CS3) | (CS4) |
| --- | --- | --- | --- | --- |
| $PM_{2.5}$ (IN)/$PM_{2.5}$ (OUT) | 0.143 | 0.470 | 0.157 | 0.020 |
| $T_i$/$CO_2$ | 0.418 | 0.122 | 0.417 | 0.256 |
| VOC/$CO_2$ | 0.303 | 0.542 | 0.456 | 0.294 |
| $PM_{2.5}$ (IN)/VOC | 0.037 | 0.062 | 0.226 | 0.480 |
| $PM_{2.5}$ (IN)/$CO_2$ | 0.074 | 0.097 | 0.226 | 0.069 |

Airtightness could also explain that $PM_{2.5}$ and VOC concentrations were only strongly associated in CS4 since both parameters were associated with indoor activities.

### 4.2. Weekly Analysis

Although average values were considered acceptable in all cases over two months, a detailed weekly analysis can change the perception of indoor air quality. This section analyzed the results from October 7 to October 14. CS1 and CS2 showed a satisfactory air renovation rate by combining infiltration and intentional ventilation, so the airtightness of these case studies does not seem to cause poor indoor air quality. Average $CO_2$, $PM_{2.5}$, and VOC concentrations were below the WHO recommendation threshold. However, Figure 6 illustrates that levels rose to dangerous values when CS2 occupants were at home.

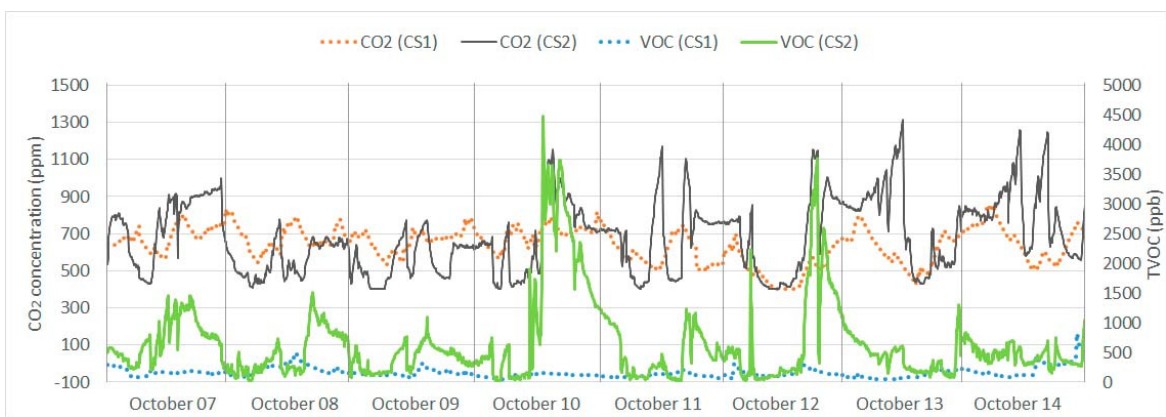

**Figure 6.** Indoor VOC and $CO_2$ concentration in New Hampshire case studies.

CS4 showed the best performance in airtightness ($ACH_{50}$ = 1.71). VOC and PM levels rose due to the increase in the house usage intensity and a likely increment in cooking and cleaning tasks. CS3 showed the worst performance. Figure 7 shows that, over the analyzed week, the average $CO_2$ concentration was 771 ppm, and 1000 ppm was the 75th percentile. These values mean that, 25% of the time, the occupants have been exposed to potentially hazardous conditions.

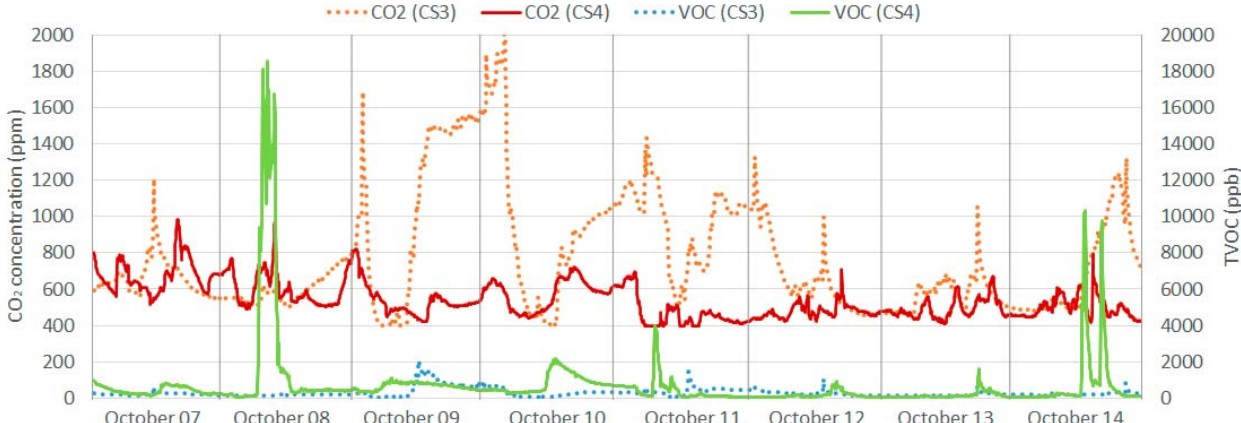

**Figure 7.** Indoor VOC and $CO_2$ concentration in Connecticut case studies.

VOC and PM risk concentrations were mainly related to continuous inhabitant presence, lack of ventilation, and occupants' unawareness about indoor air quality.

### 4.3. Daily Analysis

The daily profile of the concentrations showed a dependency on the home activities, such as cooking, working, and ventilation cycles. Figure 8 illustrates the $CO_2$ level on two different days, Thusday, 7 October and Sunday, 10 October. Dotted lines represent occupancy in the case studies, and the reported ventilation times are shown at the top of the graph. Occupants reported ventilation for a few minutes on working days in the morning, whereas only CS1 reported morning ventilation over the weekend. CS2, CS3, and CS4 reported a second period of ventilation at 11:00 a.m. (CS3) and at 4:00 p.m. (CS2, CS4). In all case studies, the concentration levels were deeply impacted by the hours of rest, with a continuous base level of indicators. CS1 and CS4, with a low occupancy per volume ratio, showed no relevant differences between working days and weekend days. In CS2, the $CO_2$ level matched the occupancy and increased above 1000 ppm in the evening when all occupants were home both days. The high infiltration rate kept the $CO_2$ levels within acceptable values.

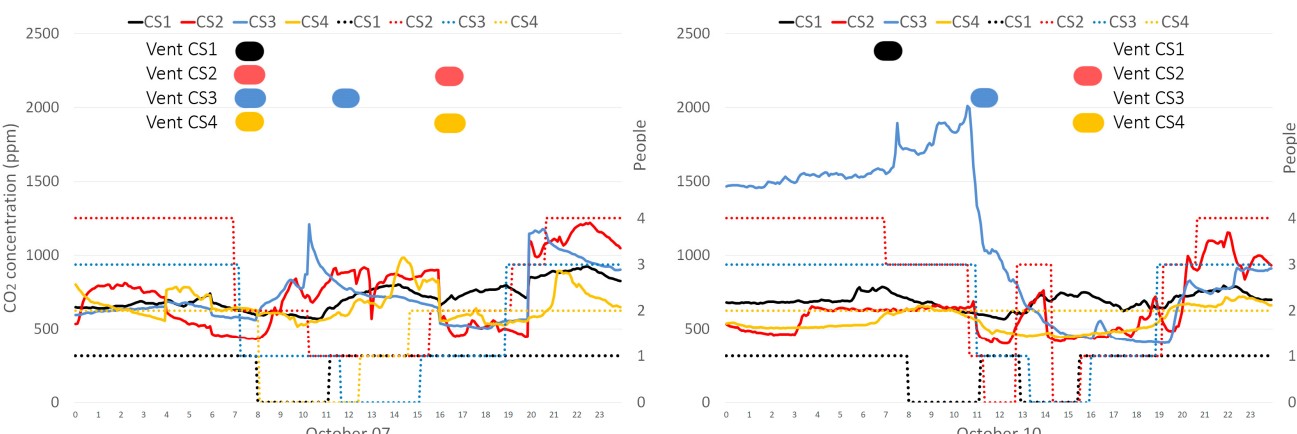

**Figure 8.** Daily indoor occupancy, ventilation, and CO2 concentration in New Hampshire (CS1, CS2) and Connecticut (CS3, CS4) case studies on Thursday, 7 October (**left**) and Sunday, 10 October (**right**).

$CO_2$ concentration in CS3 rose above 1000 ppm only at two specific moments on Thursday at 10:00 a.m. and 8:00 p.m. However, it increased up to dangerous levels on Friday afternoon, and, on Saturday morning, the levels were steadily high. By the end of the morning, it rose to 2000 ppm. The use of the fireplace over the evening and the lack of ventilation could explain the registered high $CO_2$ concentration that dropped steeply to acceptable levels when the ventilation occurred and occupants left home.

Figure 9 shows the daily profile of $PM_{2.5}$ levels and its relationship with occupancy and ventilation habits. All the studied houses were ventilated for 15 to 20 min in the morning. In CS1, CS3, and CS4, $PM_{2.5}$ concentration levels were below 50 $\mu g/m^3$. In CS2, there were peaks of 400 or even 900 $\mu g/m^3$ related to indoor activities. Working days showed peaks related to cooking activities at different times due to students' schedules renting different rooms. There was no correlation between ventilation and $PM_{2.5}$ levels.

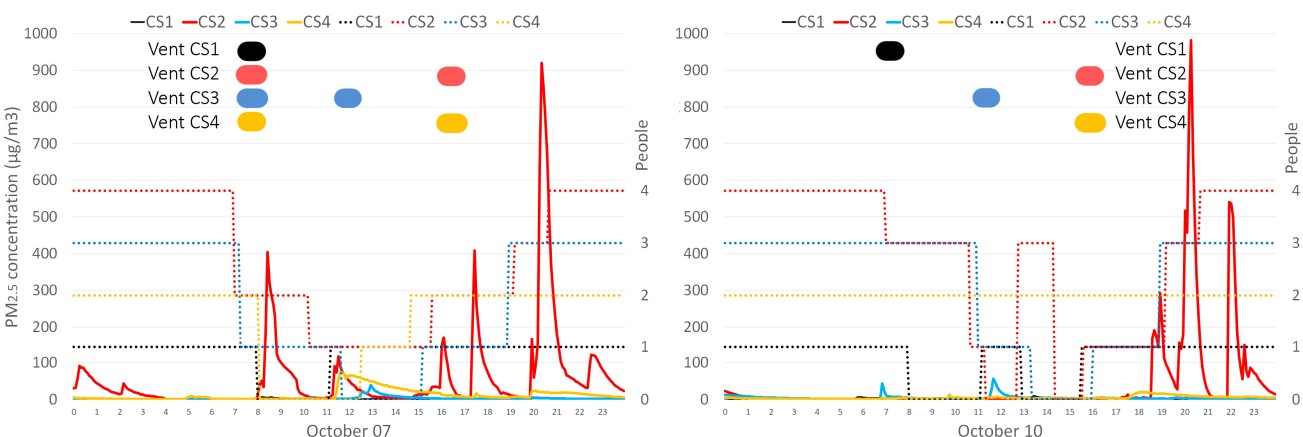

**Figure 9.** Daily indoor occupancy, ventilation, and PM2.5 concentration in New Hampshire (CS1, CS2) and Connecticut (CS3, CS4) case studies on Thursday, 7 October (**left**) and Sunday, 10 October (**right**).

Figure 10 illustrates all case studies' daily VOC concentrations on two different days. The contribution of ventilation in the dilution effect on indoor VOCs was evident, especially in CS2 and CS4. VOC concentration dropped in both cases after a few minutes of ventilation in the morning and the afternoon. However, in CS4, VOC levels rose to a dangerous concentration after cooking the dinner, and the levels remained high due to the house's airtightness.

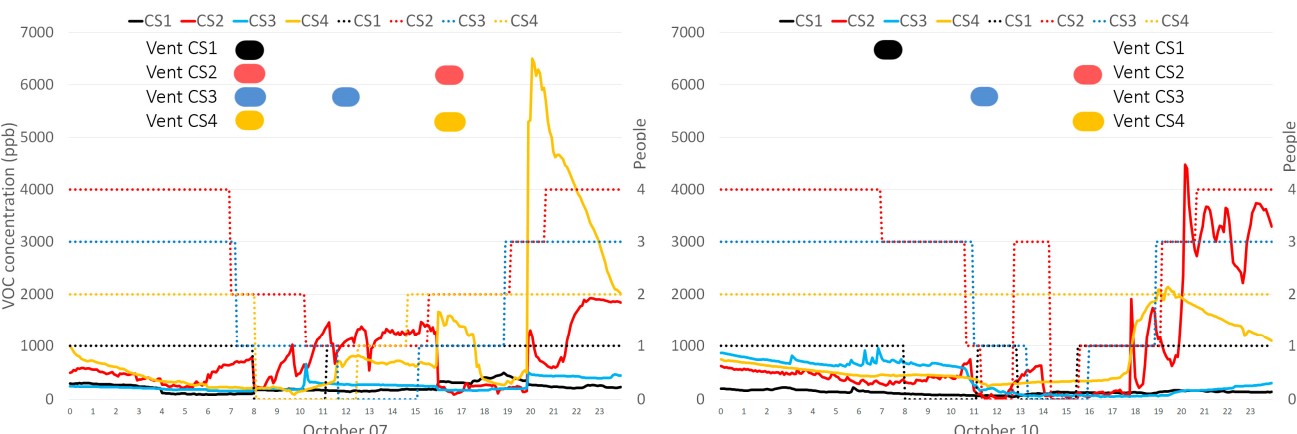

**Figure 10.** Daily indoor occupancy, ventilation, and VOC concentration in New Hampshire (CS1, CS2) and Connecticut (CS3, CS4) case studies on Thursday, 7 October (**left**) and Sunday, 10 October (**right**).

As a summary of the daily analysis, $CO_2$ levels increase with occupancy, whereas VOC levels are connected to indoor materials and activities. Besides, there is a clear correlation between the reduction in VOC and $CO_2$ concentration and ventilation habits. Most airtight case studies usually showed higher VOC concentrations, although indoor-source emission rates and indoor activities distorted the predicted patterns. Indoor $PM_{2.5}$ levels did not show a clear pattern and varied in different case studies. This indicator is connected to indoor activities so that no regular pattern can be established. This article raised the importance of replacing or improving the filtration systems of the heating and air-conditioning systems in homes. These measures must include immediate health improvements by designing efficient ventilation systems, increasing energy efficiency to prevent thermal losses, and improving the effectiveness of natural ventilation strategies.

## 5. Conclusions

Interior conditions often vary according to exterior conditions, especially seen in houses with higher leakage rates. Recently, building codes and building programs have required homes and other new constructions to be built tighter. As expected, tropospheric ozone ($O_3$), $NO_2$, $SO_2$, and $PM_{2.5}$ outdoor concentrations in this period had no meaningful impacts on indoor conditions if the airtightness of the case studies presented acceptable values (CS1, CS3, CS4). While the primary concerns regarding city outdoor pollutants are typically particulate matter and ozone over the analyzed period, outdoor conditions were somehow acceptable. Therefore, it seems possible to allocate a low influence from outdoor sources in the evolution of the indoor air quality conditions in the analyzed case studies, with the increase in internal emissions and the ventilation profiles being primarily responsible for the changes. On the other hand, there was a significant correlation, 0.47, between $PM_{2.5}$ indoors and outdoors in CS2, where the infiltration rate showed the highest value. However, this correlation showed low values in the rest of the case studies, which indicates different mechanisms of indoor emissions, such as chemicals, cleaning, and personal care products.

In CS2, occupants set heating temperatures above recommended standards during this period. In addition, conditions of lowered ventilation to reduce heating loads provoke unsuitable outdoor air quality conditions. Indoor $CO_2$ concentrations are related to ventilation strategies and airtightness. CS2 and CS3 values exceed the 1000 ppm limit recommended by the WHO for healthy environments. Over two months, the average $CO_2$ concentration was 647 ppm in CS3, and 1000 ppm was the 87th percentile. However, there were days when the average $CO_2$ concentration was 771 ppm, and 1000 ppm was the 75th percentile. According to these data, the occupants have been exposed to potentially hazardous conditions 25% of those days. CS3 was a relatively airtight dwelling with the

lowest volume per occupant, which allowed $CO_2$ concentrations to rise more than in other cases. Under these circumstances, the typical $CO_2$ emissions were above 10,000 ppb, with significant variations in concentration depending on the activities and ventilation cycles.

One of the major issues while conducting this research was the build-up of pollutants in interior spaces. Indoor particle ($PM_{2.5}$) concentration is linked with outdoor conditions. CS2 was built nearly 150 years ago when airtightness was a minor concern of homebuilders and showed the worst performance in $PM_{2.5}$ concentrations. CS2 showed values above 500 $\mu g/m^3$, whereas CS1 values remained below 50 $\mu g/m^3$ in the same area of New Hampshire.

Otherwise, VOC concentrations were affected by the irregular patterns of indoor-source emission rates linked to indoor activities. In houses without mechanical ventilation, airtightness usually provokes higher VOC concentrations. CS4 showed the highest airtightness and peak values related to cooking activities. However, this parameter is affected by different emissions and actions in the houses. CS2 showed the highest infiltration rate and the highest average concentration of VOC and $PM_{2.5}$. In cold weather, TVOC values in CS2 and CS4 usually surpassed the advised threshold of 1200 ppb. However, peak episodes above the toxicity threshold of 10,000 ppb are unusual in the homes analyzed.

Since ventilation mainly depends on envelope airtightness, the case studies with more occupants showed different values between working days and weekends. The results brought out the limitation of using $CO^2$ as the only index of indoor environmental quality. The use of other indicators showed patterns and habits that can impact occupants' health. The carbon dioxide concentration can hide other contaminants and risks.

A simple solution to poor indoor air quality would be to increase the air change rate in each house to remove pollutants efficiently. Although simple, the high costs of implementing mechanical ventilation hinder the ability to complete this measure. A more economical option would be implementing high-efficiency particulate air (HEPA) filters into heating and cooling air duct systems to trap particles indoors. Another economical option would be to provide the occupants with an app connected to the monitoring system to warn the users about low indoor air quality and convey the need for natural ventilation. Future work can include the response of the users to that app and the relationship between IAQ and energy consumption in all case studies, especially over the cold weather period.

**Author Contributions:** Conceptualization, F.d.A.G. and R.A.G.-L.; methodology, F.d.A.G. and R.A.G.-L.; formal analysis, M.G. and J.L.; data curation, R.A.G.-L., M.G., J.L. and P.Y.; writing—original draft preparation, F.d.A.G., M.G. and J.L.; writing—review and editing, M.G., J.L. and P.Y.; visualization, F.d.A.G., M.G. and J.L.; supervision, P.Y. All authors have read and agreed to the published version of the manuscript.

**Funding:** This research received no external funding.

**Institutional Review Board Statement:** Not applicable.

**Informed Consent Statement:** Informed consent was obtained from all subjects involved in the study.

**Data Availability Statement:** Not applicable.

**Acknowledgments:** This work was supported by KSC Faculty Development Grant (Keene State College, NH, USA).

**Conflicts of Interest:** The authors declare that they have no conflict of interest.

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
