# Peer review of "Assessment of Indoor Air Quality in Residential Buildings of New England through Actual Data"

_sustainability, doi:10.3390/su14020739_

Round 1

Reviewer 1 Report

This paper described a comparative survey of air quality in four residential buildings in New England, more specifically, southwestern New Hampshire and central Connecticut, including indoor temperature, RH, CO2, PM2.5, and TVOC, with plentiful and meaningful original data. Nonetheless, the objective is unclear and results expression is short of logistic, rather than the representative of four case residential building.

Concerns:

  1. What is the purpose of this study? The difference identification, the mechanism underlying, or even the influencing factors? The research target is indirect or secret with unclear expression, should be focused with more evidences.
  2. The statically analysis of the data is under poor ideas, mainly with scattered distribution line and correlation analysis. More question should be addressed with more details.
  3. The conclusion is very mint even with current plentiful data. Apart from “results revealed that 25% of a specific timeframe, the occupants have been exposed to concentration levels of CO2 above 1000 parts per million (ppm)”.
  4. The Introduction is much too complex, which could be suppressed with fewer texts.
  5. The information of Figure 1, and figure 3 are insufficient for this study with limited support. It is a better choice to place them at the supplementary materials.
  6. The titles of figures (5,6,7) are simple and without key points such as concrete duration.
  7. As to Figure 5, the intention of the current expression of Indoor temperature and CO2 concentration in New Hampshire (CS1, CS2) and Connecticut (CS3, CS4) in not clear. It is a candidate idea to list the data of each indicator from (CS1, CS2, CS3, CS4) in one figure, which would be helpful to understand the daily difference of residential quality.
  8. The total text should be improved with language polish of unnecessary information and improve the results under logistic research hypnosis.
  9. Table 3, 4, 5, and 6 are very similar and could be simplified totally. The results with significance should be marked obviously.

Reviewer 2 Report

Page 3 lines 100-103 read.... 

For each state, two residential buildings were tested trying to maintain
resemblance in the construction year and construction type. The motivation for selecting these buildings lies in the interest in comparing pollutant level differences and whether the indoor air quality standard implies differences in the energy performance..............

The features are listed in Table 1 on Page 4, and these do not match with the above statement of resemblance.

Similarly, the glazing typology/materials are not specified and the airtightness will vary from framing type. The year of construction, volume, floor area, no. of occupants, etc. vary in all four cases. 

Conclusions can be improved by addressing the above points.

Reviewer 3 Report

Some revisions according to the attached report are needed. Best regards.

Round 2

Reviewer 1 Report

All the concerns were focused and adressed with more improvements.